# From Gene to Transcript and Peptide: A Deep Overview on Non-Specific Lipid Transfer Proteins (nsLTPs)

**DOI:** 10.3390/antibiotics12050939

**Published:** 2023-05-21

**Authors:** Carlos André dos Santos-Silva, José Ribamar Costa Ferreira-Neto, Vinícius Costa Amador, João Pacífico Bezerra-Neto, Lívia Maria Batista Vilela, Eliseu Binneck, Mireli de Santana Rêgo, Manassés Daniel da Silva, Ana Luiza Trajano Mangueira de Melo, Rahisa Helena da Silva, Ana Maria Benko-Iseppon

**Affiliations:** 1Institute for Maternal and Child Health–IRCCS Burlo Garofolo, 34137 Trieste, Italy; carlos.ssilva@cesmac.edu.br; 2Departamento de Genética, Centro de Biociências, Universidade Federal de Pernambuco, Recife 50670-901, Brazil; joseribamar.ferreiraneto@ufpe.br (J.R.C.F.-N.); vinicius.amador@ufpe.br (V.C.A.); livia.vilela@ufpe.br (L.M.B.V.); mireli.santana@ufpe.br (M.d.S.R.); manasses.dsilva@ufpe.br (M.D.d.S.); analuiza.melo@ufpe.br (A.L.T.M.d.M.); rahisa.silva@ufpe.br (R.H.d.S.); 3Instituto de Ciências Biológicas, Universidade de Pernambuco, Recife 50100-010, Brazil; pacifico.joao@upe.br; 4Empresa Brasileira de Pesquisa Agropecuária, Embrapa Soja, Londrina 86085-981, Brazil; eliseu.binneck@embrapa.br

**Keywords:** genomics, expansion mechanisms, last common ancestor, plant stress, gene expression

## Abstract

Non-specific lipid transfer proteins (nsLTPs) stand out among plant-specific peptide superfamilies due to their multifaceted roles in plant molecular physiology and development, including their protective functions against pathogens. These antimicrobial agents have demonstrated remarkable efficacy against bacterial and fungal pathogens. The discovery of plant-originated, cysteine-rich antimicrobial peptides such as nsLTPs has paved the way for exploring the mentioned organisms as potential biofactories for synthesizing antimicrobial compounds. Recently, nsLTPs have been the focus of a plethora of research and reviews, providing a functional overview of their potential activity. The present work compiles relevant information on nsLTP omics and evolution, and it adds meta-analysis of nsLTPs, including: (1) genome-wide mining in 12 plant genomes not studied before; (2) latest common ancestor analysis (LCA) and expansion mechanisms; (3) structural proteomics, scrutinizing nsLTPs’ three-dimensional structure/physicochemical characteristics in the context of nsLTP classification; and (4) broad nsLTP spatiotemporal transcriptional analysis using soybean as a study case. Combining a critical review with original results, we aim to integrate high-quality information in a single source to clarify unexplored aspects of this important gene/peptide family.

## 1. Introduction

The non-specific lipid transfer proteins (nsLTPs) are a plant-specific superfamily of cysteine-rich AMPs (antimicrobial peptides). They received this name due to their ability to bind to several hydrophobic molecules, such as phospholipids and fatty acids, among others. nsLTPs are characterized by their reduced size (6.5–10.5 kDa) and the presence of eight cysteine residues (8CM domain), which form four disulfide bonds [1]. They are associated with various plant biological processes, such as growth and development, abiotic stress responses, besides plant defense [2,3,4,5,6,7,8]. The mentioned plant-originated AMPs have demonstrated remarkable efficacy against bacterial and fungal pathogens [9]. The discovery of cysteine-rich AMPs such as nsLTPs has paved the way for exploring plants as potential biofactories for synthesizing antimicrobial compounds, which holds significant promise for their application as biotherapeutic agents in the field of antimicrobial drug development [9]. It has been proposed that such antimicrobial activity is due to the nsLTPs’ ability to disrupt the permeability and integrity of the pathogens’ outer membranes, similar to other plant AMPs [7,10]. However, further studies are necessary to understand all their biological roles.

Previous works indicate that nsLTPs are encoded by a large gene family, presenting more than 50 loci in many angiosperm genomes and up to 50 loci in bryophytes, ferns, and gymnosperms [1]. Some classification systems are available for nsLTPs. These, however, are heterogeneous in terms of subgroups’ numbers and nomenclature. The classification initially proposed for nsLTPs—division into ‘nsLTP1’ and ‘nsLTP2’ subfamilies—was based on their molecular mass, sequence identity (<30% of similarity between nsLTP1s and nsLTP2s), and lipid transfer efficiency [11,12]. Disulfide bond patterns in nsLTPs may also differ. ‘nsLTP1s’ displays ‘Cys1-Cys6, Cys2-Cys3, Cys4-Cys7, and Cys5-Cys8’ pattern [13], whereas ‘nsLTP2s’ exhibit that of ‘Cys1-Cys5, Cys2-Cys3, Cys4-Cys7, and Cys6-Cys8’ [14]. Such differences in disulfide bond configuration play a crucial role in the stability of these proteins, constraining their conformational dynamics [1].

Another notable difference regards the hydrophobic cavity. The subgroup nsLTP1s may have a long tunnel-like cavity [15,16], while nsLTP2s may have two adjacent hydrophobic cavities [14]. However, it is worth mentioning that there is no specific rule in this regard since the hydrophobic cavities can vary according to the sequence of amino acid residues, the number of disulfide bonds present in the structure, and the lipid binding specificity.

Boutrot et al. [17] proposed a different classification approach. These authors introduced phylogenetic grouping as a key classification criterion. The proposition includes early diverging nsLTP homologs found in mosses and liver plants. The mentioned classification established nine groups (types ‘1’ to ‘9’) stratified with their respective consensus cysteine motifs (8CM domain)—Cys-Xn-Cys-Xn-CysCys-Xn-CysXCys-Xn-Cys-Xn-Cys (X: different amino acids; n: variable number of amino acids)—and the inter-cysteine amino acid residues diversity.

Later, another proposition, by Edstam et al. [1], sought to break the limitations inherent in the sequence conservation, considering post-translational modifications in the glycosylphosphatidylinositol (GPI) anchoring sites, intron positions, and spacing in the regions between cysteine residues, besides sequence similarity. This classification system suggests five major groups (‘LTP1’, ‘LTP2’, ‘LTPc’, ‘LTPd’, and ‘LTPg’) and four minor groups (‘LTPe’, ‘LTPf’, ‘LTPh’, ‘LTPj’, and ‘LTPk’). Despite the efforts made on the new nsLTP classification systems, the conventional classification of ‘LTP1’ and ‘LTP2’ is still widely used due to a lack of consensus among different studies. Since the nsLTP gene family is so complex and diversified, no established classification guidelines are final [18].

Most publications on nsLTP studies focus on their structure, abundance, and diversity in well-studied plant clades or model organisms. In the present work, we focused on the nsLTPs’ genome-wide annotation in some plant genomes with no nsLTP data or new genome versions available. Additionally, this work covers relevant information about the referred peptide group. It is divided into four sections, with the first three covering omics studies (genomics and transcriptomics), evolution (gene expansion mechanisms and lowest common ancestor analysis), and structural proteomics (focusing on three-dimensional structure/physicochemical characteristics) in the context of their classification.

We also present a case study for a comprehensive transcriptional analysis in soybean, considering both baseline (covering different soybean tissues and developmental stages) and differential gene expressions under diverse biotic and abiotic stresses. The present work, thus, provides a critical review, together with a meta-analysis and original results, aiming to synthesize, in a single source, high-quality information about this important peptide group.

## 2. Omics Studies for nsLTPs

### 2.1. Understanding nsLTPs from Previous Studies

Previous studies (Table 1) involving nsLTP gene search and plant transcriptional expression analyses were scrutinized (mining methodology in Appendix A). Based on this search, different views on nsLTP abundance, types (Table 1), and functions were found. The seminal work by Edstam et al. [1] proposes a greater nsLTP abundance in terrestrial plants and their absence in green algae (chlorophytes and charophytes) (Table 1), suggesting that nsLTP genes evolved soon after the terrestrial environment conquest. In favor of the mentioned proposition, a limited number of representatives and types of nsLTPs are observed when comparing lower plants (as bryophytes and lichens) to spermatophytes, which indicates the emergence of new types of nsLTPs in higher plants [1,19,20].

In the barley (*Hordeum vulgare*) genome, 70 HvnsLTPs (*Hordeum vulgare* nsLTPs) were identified (Table 1), which were classified into five groups (‘1’, ‘2’, ‘C’, ‘D’, and ‘G’) [27]. Each of these genes shared common structures. Considering their expansion mechanisms, the 70 HvnsLTP genes presented 15 tandem duplication repeats (encompassing 36 genes). The HvnsLTPs’ baseline expression profiles in different tissues across developmental stages indicated that this group of genes might perform a variety of functions [27]. In addition, the differential expression profile indicated that HvnsLTP genes might have diverged in terms of the cis-regulatory elements of their promoters [27].

For the Solanaceae family, there are data on nsLTPs for potato (StnsLTPs, *Solanum tuberosum* nsLTPs). Li et al. [29] found 83 StnsLTP genes in potato genomes, categorized into eight types, namely, ‘1’, ‘2’, ‘4’, ‘5’, ‘7’, ‘8’, ‘12*’, and ‘13*’ (Table 1). Chromosome distribution and collinearity analyses suggested that the expansion of the StnsLTP gene family was enhanced by tandem duplications. In turn, Ka/Ks analysis showed that 47 pairs of duplicated genes have gone through purifying selection during evolution. StnsLTP genes were expressed mainly in younger tissues. Furthermore, StnsLTPs contained a large number of stress-responsive, *cis*-acting elements in their promoter regions. These results indicated that StnsLTPs might play significant and functionally varied roles in potato plants.

In *Arachis duranensis*, Song et al. [31] discovered 64 AdnsLTPs (*Arachis duranensis* nsLTPs) genes, which were divided into six groups (‘1’, ‘2’, ‘C’, ‘D’, ‘E’, and ‘G’; Table 1), anchored over nine chromosomes. Considering the AdnsLTPs’ expansion mechanisms, the study revealed some gene clustering by tandem duplication, while other family members showed segmental duplication in several chromosomes. Following treatments with high salt (NaCl, 250 mM), PEG, low temperature (4 °C), and abscisic acid, the AdnsLTPs’ expression levels were altered. Three AdnsLTPs were linked to nematode infection resistance. The DOF and WRI1 transcription factors were suggested as potential controllers of the AdnsLTP response to nematode infection.

Fang et al. [30] found 330 TansLTPs (*Triticum aestivum* nsLTPs) genes in wheat (*T. aestivum*) (Table 1). Such a quantitative result can be considered an update of the 461 nsLTP loci found by Kouidri et al. [26] for the same species. To date, *T. aestivum* is the plant with the highest number of nsLTPs. The TansLTPs clustered into five groups (‘1’, ‘2’, ‘C’, ‘D’, and ‘G’) by phenetic analysis (Table 1). Gene structure and MEME pattern analyses showed that different groups of nsLTPs had similar structural compositions. Chromosome anchoring revealed that all five groups were distributed on 21 chromosomes. Furthermore, 31 gene clusters were identified as tandem duplications, and 208 gene pairs were identified as segmental duplications. Data mining of RNA-seq libraries, covering multiple stress conditions, showed that the transcript levels of some of the nsLTP genes could be strongly up-regulated by drought and high salt (NaCl, 250 mM) stresses.

In another context, Liang et al. [35] scrutinized the *Brassica napus* pangenome for BnnsLTPs (*B. napus* nsLTPs). These authors identified 246 BnnsLTP genes, divided into five groups (‘1’, ‘2’, ‘C’, ‘D’, and ‘G’; Table 1). Different BnnLTP genes were identified among the eight studied *B. napus* varieties (ZS11, Gangan, Zheyou7, Shengli, Tapidor, Quinta, Westar, and No2127). BnnsLTPs showed different duplication patterns in different varieties. Cis-regulatory elements that respond to biotic and abiotic stresses were anchored at all BnnsLTP genes. Finally, RNA-Seq analysis showed that the BnnsLTP genes were involved in responses to the fungus *Sclerotinia sclerotiorum* infection.

Vangelisti et al. [36], studying sunflower (*Helianthus annuus*) HansLTPs (*Helianthus annuus* nsLTPs), observed the existence of four (‘1’, ‘2’, ‘3’, and ‘4’) groups (Table 1). The authors did not explicitly classify the observed groups according to the available classification systems. The HansLTPs (101 in total) were further examined by looking into potential gene duplication sources, which revealed a high prevalence of tandem- in addition to whole-genome duplication (WGD) events. This finding is consistent with polyploidization events that occurred during the evolution of the sunflower genome. Three (‘1’, ‘3’, and ‘4’) of the four HansLTP groups responded uniquely to environmental cues, including auxin, abscisic acid, and the saline environment. Interestingly, sunflower seeds were the only source of expression for HansLTP group ‘2’ genes.

In line with the reports mentioned above and other works in Table 1, it is observed that the nsLTP genes act multifunctionally and show genetic variability even within accessions of the same species. nsLTPs are present in a wide range of plants, showing gene expression in different tissues, developmental stages, and stressful conditions.

### 2.2. Filling the Gap: Discovering and Classifying nsLTPs in New Plant Genomes

To provide genomic information for nsLTPs in plants not yet studied in the previous topic, and to update nsLTP data for some species with improved genome versions that have been made available, the following plant genomes were scrutinized (Table 2): (1) *Marchantia polymorpha;* (2) *Ceratopteris richardii;* (3) *Selaginella moellendorffii;* (4) *Thuja plicata;* (5) *Gossypium hirsutum;* (6) *Lactuca sativa;* (7) *Manihot esculenta;* (8) *Mimulus guttatus;* (9) *Populus trichocarpa;* (10) *Sinapis alba;* (11) *Solanum tuberosum;* and (12) *Spinacea oleracea*. The mentioned species were chosen to diversify the number of analyzed clades.

Three distinct and complementary strategies were used to retrieve nsLTPs in the genomes selected (information about the applied methodologies in Appendix A). The nsLTP exhaustive mining applied to 12 evaluated genomes (Table 2) returned 258 candidate sequences by BLASTp search, 344 by the cysteine pattern-based strategy (RegEx mining), and 1191 by the machine learning approach (HMMER tool).

The machine learning approach (HMMER tool) recovered a more comprehensive number of sequences considering nsLTP domains. Strategies based on machine learning are emerging as the future of DNA/RNA/protein sequence identification and bioinformatics, in general. The search with the RegEx approach was more restrictive, on the other hand, and not as accurate as the local alignment (BLASTp) method when observing the presence of the conserved nsLTP domain. However, it is worth mentioning that the BLASTp strategy did not recover sequences from the hypothetical proteomes of *C. richardii*, *S. moellendorffii*, and *M. polymorpha* (Table 2). When working with sequences as diverse as nsLTPs, it is advisable to combine several mining methods to increase the chances of finding the maximum number of sequences combined with subsequent data curation. In the present work, however, almost all of the sequences retrieved by the BLASTp and RegEx approaches were also retrieved by the machine-learning-based strategy .

From the analyzed species pool (Table 2), *S. moellendorffii* (21) and *G. hirsutum* (218) presented, respectively, the lowest and highest number of nsLTPs. Our search confirmed the tendency of these peptides to be encoded by large gene families: nine (75%) of the 12 analyzed species had more than 50 nsLTP loci in their respective genomes (Table 2). There was no correlation (r = 0.10; methodology in Appendix A) between genome size and the number of nsLTPs in the analyzed species pool (Table 2). However, angiosperms and the analyzed gymnosperm have a higher amount of nsLTPs than pteridophytes and the analyzed bryophyte (Table 2). This fact may be associated with the evolution of these basal groups, as shown by Edstam et al. [1]. Pteridophytes and bryophytes are phylogenetically closer to green algae than the other clades analyzed. This scenario is possibly responsible for the nsLTPs’ reduced number in the mentioned clades since no nsLTPs were identified in green algae [1].

The present work also updated the data for the nsLTP content in cotton (*G. hirsutum*), potato (*S. tuberosum*), common liverwort (*M. polymorpha*), and spikemoss (*S moellendorffii*) from the respective updated genome versions (Ghirsutum_527_v2.1, Stuberosum_686_v6.1, Mpolymorpha_320_v3.1, and Smoellendorffii_91_v1.0 | Phytozome database). While Li et al. [6] and Li et al. [29] identified, respectively, 91 and 83 nsLTP loci for the *G. hirsutum* and *S. tuberosum* (Table 1), 218 and 105 nsLTP loci were identified in our data mining approach (Table 2). For *M. polymorpha* and *S. moellendorffii*, 21 and 36 nsLTPs were found, respectively (Table 2), in the present study (compared to 13 and 23 nsLTPs reported by Fonseca-García et al. [34]; (Table 1)). Genome assemblies are never perfect since they are models for the actual genome. It is hard to completely rule out all potential technological or algorithmic flaws, and no single assembly can accurately capture all the variety within populations of a species. Thus, published genomes that have an active research community are continuously improved. Such modified and updated versions are potential sources for changing minor paradigms.

Considering the 1191 nsLTPs’ categorization (nsLTP domains are available in Appendix A), we observed nine large groups, of which five could be identified (‘nsLTL1’, ‘nsLTL2’, ‘nsLTLG’, ‘nsLTLD’, and ‘nsLTLC’; Figure 1), in addition to four distinct groups denominated ‘Unknown 1-4’ (Table 2). The complete tree, with bootstrap values and identified sequences with assigned categories, is available in Appendix A.

Group separation, obtained from the implemented neighbor-joining (NJ) approach (methodology in Appendix A), reprised the nine groups in the Edstam et al. [1] classification. The strategy of performing NJ analysis from 8CM domain sequences (as performed by Edstam et al. [1] and Xue et al. [38]) promoted better group discrimination. The NJ tree derived from the complete nsLTP sequences (8CM + upstream and downstream regions) did not present such a level of resolution (only six groups were formed.

As will be seen in the ‘How structural nsLTP proteomics correlates with current nsLTP classification systems?’ section, nsLTP sequences show high variability in the amino acid sequence. Outside the 8CM region, the mentioned variability is accentuated, which causes greater noise in the distance analysis, resulting in an efficiency reduction in the formation of ‘true’ groups. Although reduced, compared to other nsLTP regions, the variability of the 8CM region was also a significant factor in the analysis using the NJ method. This is evidenced by the low bootstrap values of the first branches formed (Appendix A) and in nsLTP NJ analysis for a manifold of species (see works by Fang et al. [30] and Vangelisti et al. [36], among others, in Table 1). Bootstrap values reflect the proportion of trees/replicates in which a recovered grouping is presented (in other words, a measure of support for that group). Despite reduced bootstrap values, the obtained tree topology was in accordance with the composition of the characterized seed sequences (methodology in Appendix A) used to perform the nsLTP classification.

Regarding the nsLTPs’ composition in the scrutinized species, *L. sativa*, despite not having the highest amount of nsLTPs in its genome (Table 2 and Table 3), was the species which presented the greatest variety of these peptides (Table 3), with at least one member of each of the nine groups found. *M. polymorpha*, in turn, had the lowest variety of nsLTPs (presenting members only for the ‘LTP1’, ‘LTP2’, ‘LTPD’, ‘LTPG’, ‘Unknown 1’, and ‘Unknown 4’ groups; Table 3), a fact that is possibly associated with its nsLTPs’ small genomic amount. Our results for this species, however, indicated that it presented nsLTP groups (e.g., ‘LTP1’, ‘LTP2’, ‘Unknown 1’, and ‘Unknown 4’) not yet identified in previous studies (i.e., in Edstam et al. [1] and Fonseca-García et al. [34]).

In another context, considering the different groups, ‘nsLTPG’ was present in the 12 analyzed plant species, while the ‘Unknown 4’ group was specific to four species (*C. richardii*, *L. sativa*, *M. polymorpha*, and *S. moellendorffii;*
Table 3; Appendix A). Subgroups were formed for ‘nsLTPG’ and ‘nsLTPD’ (Figure 1; Appendix A). In the case of ‘Unknown 1-4’ (Figure 1; Appendix A), no classification could be assigned based on the set of used seed sequences. The topological organization and sequence composition of the ‘Unknown’ groups needs further investigation.

## 3. ‘nsLTP Evolution’ Section

### 3.1. The Landscape of the nsLTP Expansion in Plant Genomes

Gene duplication is a crucial evolutionary mechanism providing new genetic material to a genome. New proteins often arise from duplicated copies generating unprecedented capabilities by diversifying protein functionality (paralogy). We analyzed and categorized such mechanisms (methodology in Appendix A) for the 12 species presented in Table 1. In addition, soybean (*G. max*) was also analyzed in this context. Although already studied in terms of nsLTP mining and classification [34], this species does not present information about nsLTPs’ expansion mechanisms.

Singleton nsLTPs were a minority in the analyzed genomes (Figure 2), indicating a high duplication rate of loci associated with this gene family. In the present study, two main mechanisms with different performance levels were identified as responsible for nsLTP expansion (Figure 2): dispersed and tandem duplications. Despite different duplication mechanisms, it is clear that similar evolutionary forces acted in the nsLTPs’ expansion in these different genomes.

Dispersed duplicated genes are prevalent in different plant genomes [19]. They generate paralogs that are neither near each other on chromosomes nor show conserved synteny [39]. This is in line with the random nature of the distribution of the analyzed loci in their respective anchor species.

Regarding the second most active mechanism, gene tandem duplication has been commonly found to be important for plant adaptation to rapidly changing environments. In contrast, genes expanded by non-tandem mechanisms tend to have intracellular regulatory roles [40]. Thus, the set of duplication mechanisms observed (mostly tandem and dispersed) may be associated with the heterogeneous physiological role that nsLTPs play in plants. The mentioned peptide group plays roles ranging from plant defense (for a review, see Liu et al. [8]) to liquid secretion, cuticular wax accumulation, pollen, and seed development, among other functions [20].

Considering the genesis of the distribution pattern, the dispersed one would be more expected in polyploids than in diploids. However, this assumption was not confirmed when comparing the two pteridophytes since there was a significant prevalence of dispersed distribution in *S. moellendorffii* (diploid with 2n = 16), while in the polyploid *C. richardii* (2n = 78), the two categories dispersed (44) and tandem (41) had similar prevalence (Figure 2).

In the diploid gymnosperm *T. plicata* (2n = 22), the species with the largest genome among those analyzed (ca. 12.5 Gb), the tandem dispersal mechanism clearly prevailed (Figure 2), indicating a possible importance of segmental duplications in the genesis of the high number of nsLTPs in this species.

Concerning the angiosperms (Figure 2), the species with the highest number of nsLTPs (218) was cotton (*G. hirsutum*, 2n = 52) with a prevalence of the dispersed mechanism, indicating the importance of WGD (i.e., polyploidy) in the expansion of nsLTPs. A similar result was observed in soybean (*G. max*), also polyploid (2n = 40). In contrast, this trend was not observed in potato (*S. tuberosum*), also considered polyploid (2n = 48), where a prevalence of the tandem (50 genes) mechanism was observed, followed by the dispersed mechanism (38). Considering the above, the mechanism of distribution and expansion of nsLTPs (and other gene families) needs to be carefully analyzed for each taxon and lineage without generalizations. This is especially true considering that some polyploids undergo diploidization processes with the retention of duplicate copies of some regions and the elimination of other ones (e.g., Renny-Byfield et al. [41]).

Future actions, such as the transcriptional expression study of tandem duplicated loci, will add more information about the evolutionary impact of identified duplication events. According to Liu et al. [8], the duplicated nsLTP genes might keep certain crucial activities in the course of subsequent evolution. The identical expression patterns could be explained by the duplicated genes’ remarkably similar protein architecture. In turn, duplicated genes with a considerable variation in expression may result from duplication events with considerable alteration of gene regulation.

### 3.2. nsLTP Lowest Common Ancestor (LCA) Analysis

Understanding the evolution, diversity, and abundance of nsLTPs within plants is not a trivial task. On this point, evaluating available genomic data using bioinformatics approaches may help the nsLTP origin and diversification studies. Here, we use LCA analysis (methodology in Appendix A) for the first time with nsLTPs, generating an nsLTP phylogenetic tree according to a taxonomic rank obtained from the NCBI Taxonomy database (Figure 3) using the Edstam et al. [1] classification. The LCA approach corresponds to an evolutionary method based on orthologous groups, part of Ukkonen’s Algorithm to construct suffix trees, aiming to determine the common ancestor between two or more distinct organisms based on a taxonomic tree [42,43]. The branching pattern in a phylogenetic tree (Figure 3) reflects how species or other taxonomic groups evolved from a series of common ancestors.

The nsLTP sequences used to generate the phylogenetic tree were grouped in different taxonomic categories, including Magnoliophyta (914 matches), Lycopodiopsida (five matches), and Bryopsida (one match) for the Viridiplantae clade (Figure 3). The LCA analysis confirmed the presence of nsLTP classes in plants, with some of them (e.g., LTP1 and LTP2) widely distributed among higher plants (Figure 3), corroborating the works by Edstam et al. [1], Salminen et al. [20], and Missaoui et al. [44]. Additionally, no green algae were observed in our LCA tree, strengthening the hypothesis of nsLTP origin in land plants about 470 million years ago [1].

The scientific literature mentions that higher plants exhibit exclusive nsLTPs (with different cysteine patterns) when compared to non-vascular plants (such as the moss *Physcomitrella patens*). These nsLTPs show low sequence similarity to the nsLTP genes of non-vascular plants [1]. From this presumption, one question arises: why do higher plants present exclusive classes not present in primitive ones? Most of these nsLTP classes probably result from evolution during terrestrial environment occupation, likely linked to their role in stress adaptation and defense, as proposed by Edstam et al. [1] and Liu et al. [8].

The earliest nsLTP groups, found in all land plants, seem to be members of nsLTPd and nsLTPg, presenting probably a common origin. Furthermore, the LCA tree revealed that nsLTPs from some species tend to cluster together—based on their structural features—with members of other distinct taxa (for more information, see the ‘Structural proteomics’ section), especially when considering early diverging angiosperms [26,34,44]. This could indicate that duplication events in the nsLTP family occurred independently in the different plant groups, even at the intraspecific level, leading to multi-functionalization of nsLTPs. As reviewed by Amador et al. [7], this great diversity and apparent redundancy of nsLTPs in angiosperm genomes, and the neofunctionalization process, contribute to paralogs’ emergence and their multiplicity of functions.

The molecular evolution of plant nsLTPs evidenced by LCA (Figure 3) showed that nsLTP diversification was mainly based on gene duplications, a known driving evolutionary force and an important factor in increasing genome complexity. Duplication events are probably the main mechanism of nsLTP evolution, including tandem and dispersed duplications (as discussed in the ‘The landscape of the nsLTP expansion in plants’ topic). In addition to the results presented here, the works by Liang et al. [35] and Yang et al. [45], among others, also corroborate this statement.

The presence/occurrence of horizontal gene transfer to/from bacteria should also be addressed (Figure 3). Appendix A and Figure 3 point to nsLTP diversification based on multiple duplication events in Viridiplantae, possibly associated with terrestrial environment occupation by land plants, and it was directly associated with leaf cuticle formation to avoid dehydration. Possible duplication events are indicated by black dots (Figure 3 and Appendix A), pointing out shared nsLTP subgroups among related taxa. These duplication events are thought to have affected the nsLTP gene family, allowing both conservation and divergence of gene function [27,44]. Our LCA analysis, using plant nsLTPs as seeds in alignment against RefSeq-NCBI, also identified a few proteins from bacteria (Gammaproteobacteria (28 matches) and Bacilli (one match)) as well as Metazoan (Mammalia (one match); Figure 3) that contained the nsLTP motif and domain. The mammalian protein sequence identified (accession number XP_036992713, Appendix A) regards the Jamaican fruit-eating bat (*Artibeus jamaicensis*), showing high similarity (87.78%) to a plant nsLTP1 amino acid sequence. Since no other nsLTP genes were identified for Metazoans, we believe this protein sequence may be a plant contamination of the genomic Metazoan sequencing data. We also obtained hits for two different species of bacteria [*Acinetobacter baumannii* (WP_143045904; Appendix A) and *Paenibacillus* sp. (WP_083442884; Appendix A)], presenting similarity to true non-plant nsLTP proteins and conserved domains related to nsLTPs from plants. Here, we consider this discovery as a possible case of HGT (horizontal gene transfer) since HGT is a well-known and pervasive evolutionary mechanism in prokaryotes for different gene families that could improve the adaptive ability of prokaryotes in changing environments [29,46]. Anyway, contamination cannot be discarded in all situations. Therefore, the presence of such sequences homologous to plant nsLTPs demands further investigation and confirmation.

Our LCA results for nsLTPs do not indicate that the complete nsLTP domain is present in proteins from species outside the plant kingdom. It is clear that to elucidate all questions regarding this kind of ‘evolutionary history noise’ of the nsLTP gene family depends on a broader range of sequenced species deposited on databases that will contribute to future studies to elucidate nsLTP functions and relevant mechanisms associated with their evolutionary origin and diversification, even considering HGT events.

## 4. ‘Structural Proteomics’ Section

### How Structural nsLTP Proteomics Correlate with Current nsLTP Classification Systems?

Although some nsLTPs’ three-dimensional structures have been determined in different plant species, most structural studies are limited to nsLTPs of types ‘1’ and ‘2’ [44,47,48]. In this analytical context, the other nsLTP types are still poorly addressed. To circumvent this problem, this work generated theoretical protein models for nsLTP sequences (methodology in Appendix A) employed in two landmark studies [1,17] and widely used as current classification systems for this protein group. Forty-nine theoretical models for nsLTPs provided (Appendix A) and classified (i.e., types ‘1’ to ‘9’, in addition to type ‘Y’) by Boutrot et al. [17] were obtained. All these referential models are shown in Figure 4A,B. Additionally, another 41 models were generated for nsLTPs provided (Appendix A) and classified (i.e., ‘LTP1’, ‘LTP2’, ‘LTPc’, ‘LTPd’, ‘LTPg’, ‘LTPe’, ‘LTPf’, ‘LTPh’, ‘LTPj’, and ‘LTPk’) by Edstam et al. [1]. All these referential models are available in Appendix A.

Besides studying the nsLTP structures, we aimed to analyze protein modeling data to test whether resemblances/differences in the three-dimensional conformation and physicochemical parameters (physicochemical variables in Appendix A) result in clusters similar to those obtained with the parameters adopted by Boutrot et al. [17] or Edstam et al. [1] in their classification systems.

In general, for both analyzed works, it was observed that the three-dimensional structures of the different nsLTP types/groups were similar (Figure 4A; Appendix A). Exceptions, however, were found for the theoretical models of type ‘3’ nsLTPs (Boutrot et al. [17] classification; Figure 4A) and the nsLTPc group (Edstam et al. [1] classification), which presented a different structural configuration, since that they have only two disulfide bonds (‘Cys1-Cys5’, ‘Cys2-Cys3’). These structures are mainly composed of α-helices, common in nsLTPs, which can vary from three to four and are connected by small loops and an unstructured C-terminal tail (Figure 4A; Appendix A).

The position of α-helices creates an internal hydrophobic cavity (Figure 4B; Appendix A), suitable for binding lipids and other hydrophobic molecules. In general, such a cavity is a conserved feature in nsLTPs [7]. It is noteworthy that the association between hydrophobic molecules and hydrophobic pockets occurs with different affinities, depending on the molecule size and structure. However, this association does not cause any change in the protein’s three-dimensional conformation [49].

Despite the structural conservation observed in the generated models (Figure 4A; Appendix A), the nsLTPs analyzed by Boutrot et al. [17] and Edstam et al. [1] presented a high diversity of amino acid residues among the conserved cysteine sites (Appendix A). Such heterogeneity is commonly observed in nsLTPs (and other cysteine-rich peptides). It may be related to the fact that they are involved in a multitude of plant molecular mechanisms, including defense against pathogens [50].

Given the diversity in the nsLTPs’ amino acids’ linear composition, there is a need to understand the peptide ‘sequence–structure–classification’ relationship. Such information will enable the understanding of how the obtained three-dimensional structures and their physicochemical characteristics relate to the classification systems [1,17] scrutinized here.

Considering the above, when analyzing the RMSD distance, i.e., the measure of the average distance between the atoms (usually the backbone atoms) of superimposed protein models (RMSD values in Figure 4A, Appendix A, and Appendix A), it was possible to observe that most representative present similar values. The exceptions, however, were type ‘3’ nsLTPs (Boutrot et al. [17] classification) and the ‘nsLTPc’ group (Edstam et al. [1] classification). As mentioned earlier, these nsLTP classes presented a different three-dimensional structure than their peers. The almost generalized similarity was also found in the structural PCA performed with the models. The results presented in Figure 5A,B are inconsistent with the Boutrot et al. [17] and Edstam et al. [1] classification systems since they showed a generalized similar three-dimensional conformation. The explanation for the clustering pattern obtained in Figure 5A,B is related to the conservation of the structure to the detriment of the amino acids’ diversity among the cysteine residues anchored in the different nsLTP types/groups. The three-dimensional structural conservation makes it challenging to differentiate the nsLTPs into distinct and isolated groups. However, it is worth noting that there is some incipient group formation, such as type ‘3’, type ‘5’ (except for 5.1, which has five amino acid residues less in the C-terminal tail compared to the others of type ‘5’), and type ‘6’ (Figure 5A). The mentioned types slightly stand out from the other structures in the principal components (PCs) area (Figure 5A). In the structural PCA considering the sequences classified by Edstam et al. [1], however, such a small separation does not occur (Figure 5B).

In another context, when analyzing the physicochemical properties with PCA (Figure 6A,B), the absence of individualized groups was also observed. As mentioned before, the nsLTP sequences are very variable in terms of linear amino acid composition. Such diversity also results in variant physicochemical properties. In the sequence similarity index (Appendix A), groups ‘4’, ‘5’, ‘8’, ‘9’, and ‘Y’ (Boutrot et al. [17] classification) presented higher indices of intragroup similarity. Considering the classification by Edstam et al. [1], there was no group formation based on the similarity index (Appendix A).

Despite the almost general conservation of tertiary structure (Figure 4A; Appendix A), the surface charge qualitatively varied (positive or negative, but with a preponderance of positive charges in the studied sets) among the nsLTP types/groups (Figure 7 and Appendix A). In addition, the number of hydrophobic, acidic, basic, and neutral amino acid residues also was diverse. Such observations may be associated with the different nsLTP molecular roles.

Considering both PCA analyses (based on (i) physicochemical features of aa residues or (ii) the peptide tertiary structure), no correlation could be made with the classification systems proposed by Boutrot et al. [17] and Edstam et al. [1]. Even intragroup, there was significant variation in the physicochemical characteristics of the amino acid residues of nsLTPs. In turn, considerable conservation of the tertiary structure was observed (except for type 3, classification by Boutrot et al. [17], or type C, classification by Edstam et al. [1]). Even being not yet ideal, these classifications are less artificial than the initially proposed grouping [11,12] into ‘nsLTP1’ and ‘nsLTP2’ subfamilies—based mainly on molecular mass and sequence identity.

In summary, current classifications target mainly the linear structures of nsLTPs, without major implications considering their specific functions. A future association of their roles, in terms of functional omics, with the physical–chemical and tertiary characteristics uncovered here may help elaborate a less artificial classification system reflecting the function of nsLTPs.

## 5. nsLTP Transcriptional Expression: Soybean as a Case Study

### Expression in Soybean Plants: From Baseline to Differential Expression

Baseline gene expression analysis in 14 tissues/developmental stages of soybean (methodology in Appendix A) revealed that GmnsTLPs (*Glycine max* nsLTPs) were expressed on all analyzed substrates (Figure 8). Eighty-three (~70%; Appendix A) of the 120 GmnsLTP loci (Appendix A) were expressed in at least one of the analyzed substrates. About 15% (13 GmnsLTP loci) were expressed in only one or two different tissues/developmental stages, while approximately 44% (37 GmnsLTP loci) were expressed in at least 10 different tissues or developmental stages. Eighteen percent (15 GmnsLTP loci) were expressed on all analyzed treatments. GmnsLTPs were expressed mainly in flower (59 GmnsLTP loci), followed by ‘one centimeter pod’ (57) and root (57) (Figure 8). The lowest representation of expressed GmnsLTP loci occurred in ‘seed 42DAF’ (Figure 8).

Regarding the group with the highest number of expressed loci, GmnsLTPg stood out in 13 of the 14 tissues/stages studied (Figure 8), followed by the GmLTP1 group in seven, and by the GmnsLTPd group, in six tissues/stages. Interestingly, none of the analyzed tissues/stages presented expression of GmnsLTPc or GmnsLTPe representatives. The explanation for their absence remains an open question. In another context, there was no direct relationship between tissues/developmental stages and expression level of the different tested loci; it was observed, however, that there were loci with high levels of expression (RPKM counts > 100) in several tissues (Appendix A). These may become future targets for gene manipulation, with consequent functional characterization of GmnsLTPs.

Our analysis showed that nsLTP1, nsLTPg, nsLTPd, and nsLTP2 groups are expressed in all studied situations. At the loci level, however, promiscuous expression of certain GmnsLTPs (Appendix A) was found, while others were explicitly expressed only in a given tissue/stage of development. Regarding the first-mentioned expression group, such ubiquity in soybean tissues may be associated with the defense role played by these proteins. Some nsLTPs have been classified as belonging to the pathogen-related (PR) protein group, specifically, the PR14 family [51]. A broad line of evidence indicates an nsLTP role in plant defense (for review, see Liu et al. [8]). Notably, the highest amount of expressed GmnsLTP loci was found in floral, root, and young fruit (‘one centimeter pod’) tissues. Flowers have been implicated as hubs of disease transmission [52], while root tissue is a site commonly inhabited by microorganisms (whether pathogens [53] or symbionts [54]). In turn, soybean pods are also substrates affected by important diseases (as in soybean pod blight [55]).

Regarding the GmnsLTP loci with specific expression, in addition to being able to perform defense functions, such proteins may be associated with less generalist functions, such as cuticular wax accumulation, liquid secretion, pollen and seed development, and seed germination [20]. As mentioned, nsLTPs are considered critical proteins for the plant’s survival and land colonization.

Regarding the differential expression approach, four assays were analyzed (methodology in Appendix A) covering 20 different treatments (Figure 9; Appendix A). The expression of 80 distinct GmnsLTP loci were detected (Figure 9; Appendix A).

nsLTPs are often associated with defense against pathogens (e.g., Xue et al. [38]). Interestingly, there was no detection of differential expression (−1 > Log_2_FC > 1 and FDR < 0.05) after inoculation with *Fusarium oxysporum* (pathogenic and non-pathogenic isolates) in soybean root tissue (up to 96 h post-inoculation). No differential expression was observed in response to inoculation with MAMPs (flg22 and chitin) in two parental soybean lines and two progeny lines (from 1 to 8; Figure 9). Detailed analysis of these assays showed that some GmnsLTPs exhibited transcriptional modulation (Log_2_FC) with associated FDR < 0.05. However, the mentioned Log_2_FCs did not show values above the cut-off (−1 > Log_2_FC > 1) limits. Thus, a possible reason for the absence of up-regulation for assays involving biotic conditions in soybean may be associated with a methodological bias of the differential expression.

The mining of transcripts encoding nsLTP proteins in assays under abiotic conditions, however, revealed the up-regulation of several GmnsTLPs, namely: 3, for the treatment ‘dehydration stress’ at ‘12 h’ vs. ‘control’ (number 10; Figure 9); 15 for the treatment ‘salt stress’ at ‘12 h’ vs. ‘control’ (number 13; Figure 9); 8 for the treatment ‘salt stress’ at ‘6 h’ vs. ‘control’ (number 14; Figure 9); and 2, for the treatment ‘water deficit, 30% soil field capacity’ vs. ‘control, 70% soil field capacity’ at ‘Zeitgeber time 0’ (number 15; Figure 9). The mentioned amount was associated with 23 non-redundant loci (indicated by red spheres in Figure 9).

The studied data demonstrated an important association between GmnsLTPs and salt stress response, highlighting the GmnsTLPd and GmnsTLPg groups. These presented, respectively, eight and five up-regulated loci for the treatment of ‘salt stress’ at ‘12 h’ vs. ‘control’ (number 13; Figure 9), as well as four and three up-regulated loci for the treatment ‘salt stress’ at ‘6 h’ vs. ‘control’ (number 14; Figure 9). The nsTLPs’ participation in the response to saline conditions has been reported in the scientific literature. Akhiyarova et al. [56] assumed that these proteins could participate in salt-induced pea root suberization or transport of phloem lipid molecules. In turn, Xu et al. [57] observed that nsLTP enhances salt and drought tolerance in *N. tabacum*. Such information adds importance to this peptide group, reinforcing its multifunctional role in plants.

## 6. Conclusions, Perspectives, and Open Questions

There is growing interest in nsLTPs due to their multifunctional roles in plant molecular physiology, which has resulted in some comprehensive reviews regarding their structure and function. Due to this, the present work has proposed a different approach. In addition, to recover published information about the omics of the mentioned gene family, we focused on investigating topics such as evolution and structure using strategies that are still little applied for nsLTPs (as is the case with LCA, expansion mechanisms, and high-throughput protein modeling).

Our investigation corroborates previous works confirming the absence of the nsLTPs and their protein domain in species outside the plant kingdom. Sequences from other groups in databanks with similarity to nsLTPs were considered contamination artifacts (as in the case of the bat *Artibeus jamaicensis*). Concerning the sequences found in bacteria (*A. baumannii* and *Paenibacillus* sp.), they can also be contaminants, or may result from horizontal gene transfer, which needs experimental confirmation.

Tandem and dispersed duplication events figure as the main nsLTP evolution mechanisms in plants. Interestingly the number of nsLTP representatives may vary when comparing plants with similar complexity, but they do not ever correlate with the genome size or the ploidy level.

The gene expression analysis in soybean revealed a complex transcriptional regulation of these peptides, which can be transcribed in a specific or a promiscuous way (considering tissues or developmental stages). Additionally, we emphasized nsLTPs’ up-regulation in different tissues under abiotic stresses (such as high salt and drought).

An insight regarding the ‘sequence–structure–classification’ relationship, in turn, revealed a high conservation of the nsLTPs’ three-dimensional structure, in contrast with a significant variation in its physicochemical characteristics (even intragroup). It is clear that the initial classification system (dividing them into nsLTP1 and 2) is artificial. Physico-chemical variables should be associated with phenetics and functional features to generate a functional classification system for this peptide group.

Several challenges need to be overcome before we can completely comprehend the roles that nsLTPs play in plants, despite all the efforts made to uncover the diverse spectrum of functions of these molecules. We still cannot explain why nsLTPs exert diverse activities in plants despite sharing the considerable number of structural similarities uncovered here. Research into nsLTP function and evolution shall be accelerated by systems biology studies, with further downstream biochemical analysis and transgenic (or gene edition) research, which will also clarify the precise functions *in planta* of these peptides. The informational body available here enriches our comprehension of this heterogenic, plant-specific peptide family.

## Figures and Tables

**Figure 1 antibiotics-12-00939-f001:**
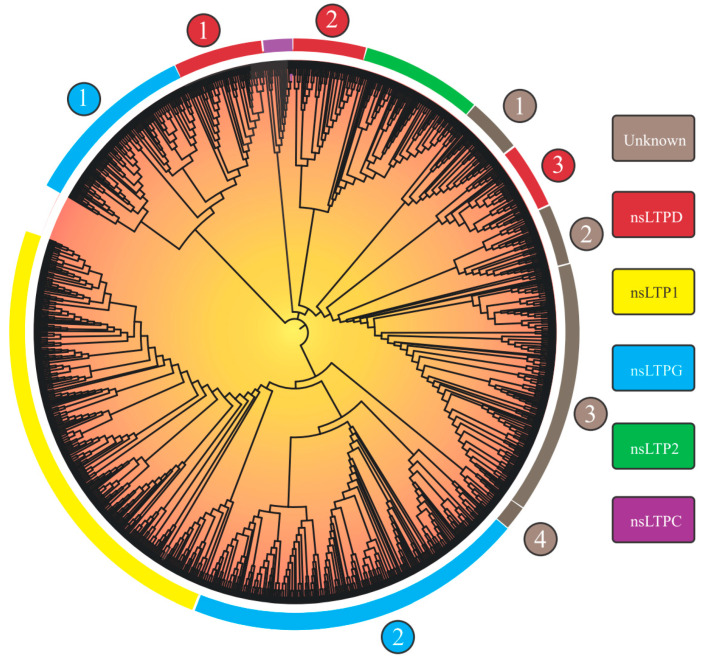
Neighbor-joining tree (based on 8CM nsLTP domains) of 1191 non-specific lipid transfer proteins (nsLTPs) predicted in *Marchantia polymorpha*, *Ceratopteris richardii*, *Selaginella moellendorffii*, *Thuja plicata*, *Glycine max*, *Gossypium hirsutum*, *Lactuca sativa*, *Manihot esculenta*, *Mimulus guttatus*, *Populus trichocarpa*, *Sinapis alba*, *Solanum tuberosum*, and *Spinacea oleracea* genomes. All amino acid sequences were aligned using ClustalX2. The obtained result was visualized with the iTOL program. Legend: the numbers (1–4) inside the circles on the edge of the tree indicate different nsLTP subgroups of a given nsLTP group; at the rectangles, the nsLTP group classification is available.

**Figure 2 antibiotics-12-00939-f002:**
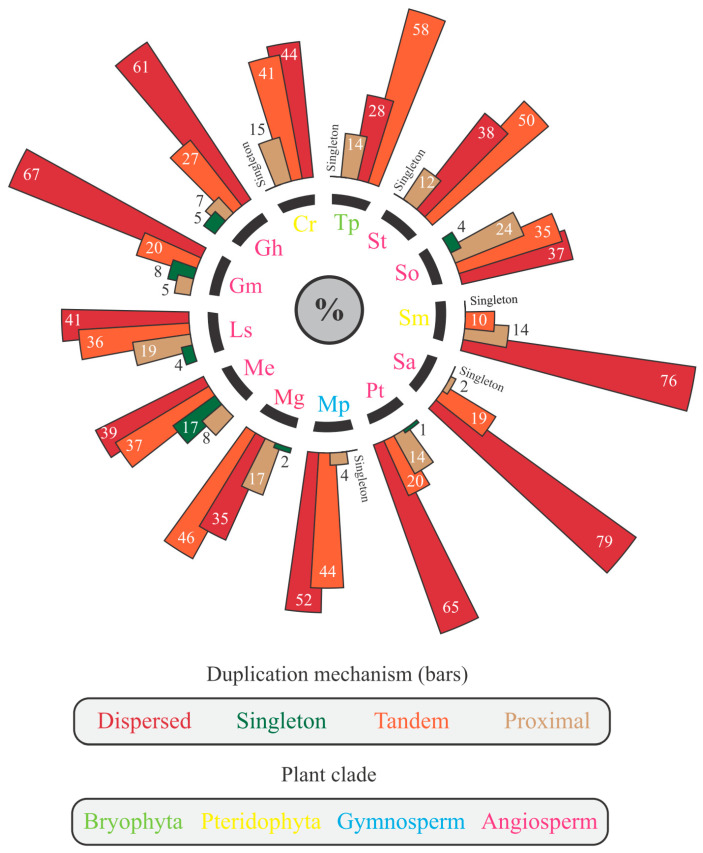
Categorization and quantification of the nsLTPs’ main genomic expansion mechanisms in Marchantia polymorpha (Mp); Ceratopteris richardii (Cr); Selaginella moellendorffii (Sm); Thuja plicata (Tp); Glycine max (Gm); Gossypium hirsutum (Gh); Lactuca sativa (Ls); Manihot esculenta (Me); Mimulus guttatus (Mg); Populus trichocarpa (Pt); Sinapis alba (Sa); Solanum tuberosum (St); and Spinacea oleracea (So) genomes. Legend: Data presented in percentage values.

**Figure 3 antibiotics-12-00939-f003:**
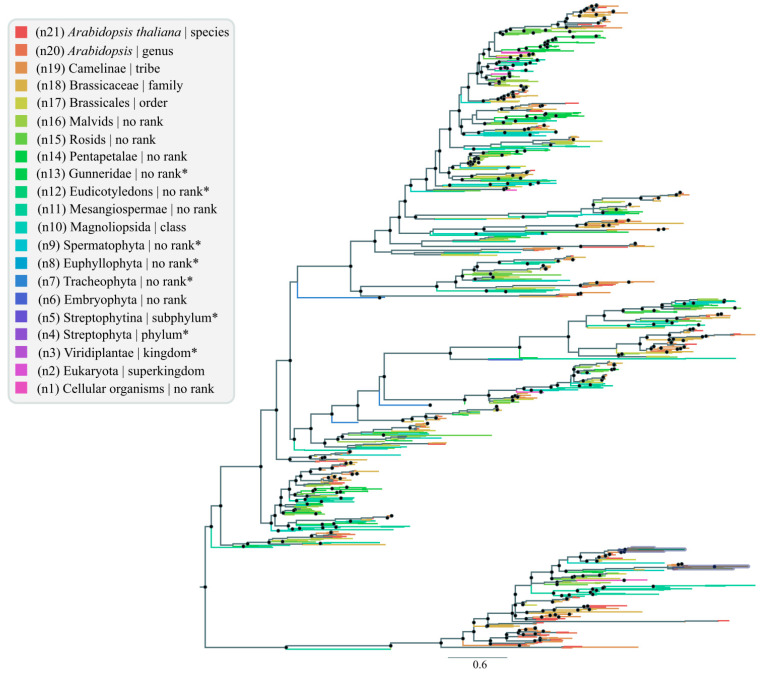
Lowest common ancestor (LCA) dendrogram, representing relationships from 950 different RefSeq-NCBI nsLTP sequences. Black dots represent points of nsLTP gene duplication. Legend: n (number): LCA levels; * LCA levels not found in the obtained tree; black dots indicate possible duplication events.

**Figure 4 antibiotics-12-00939-f004:**
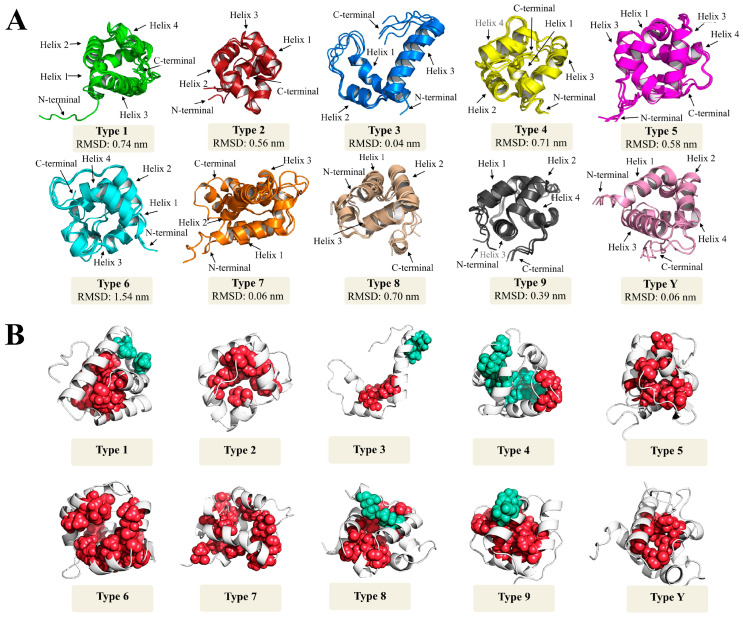
Three-dimensional reference structures for different types of nLTPs classified by Boutrot et al. [17] and visualized in Pymol. (**A**) Theoretical model alignment of different nsLTP types, solved by Alpha-Fold2, presenting their respective RMSD (root-mean-square deviation of atomic positions) values. For each given structure, black arrows indicate the location of the N- and C-terminal sites and α-helices. (**B**) Hydrophobic clusters, analyzed by ProteinTools, relative to the theoretical three-dimensional reference models of the different nsLTP types classified by Boutrot et al. [17]. The green and red spheres represent different clusters formed from interactions between nearby amino acid residues.

**Figure 5 antibiotics-12-00939-f005:**
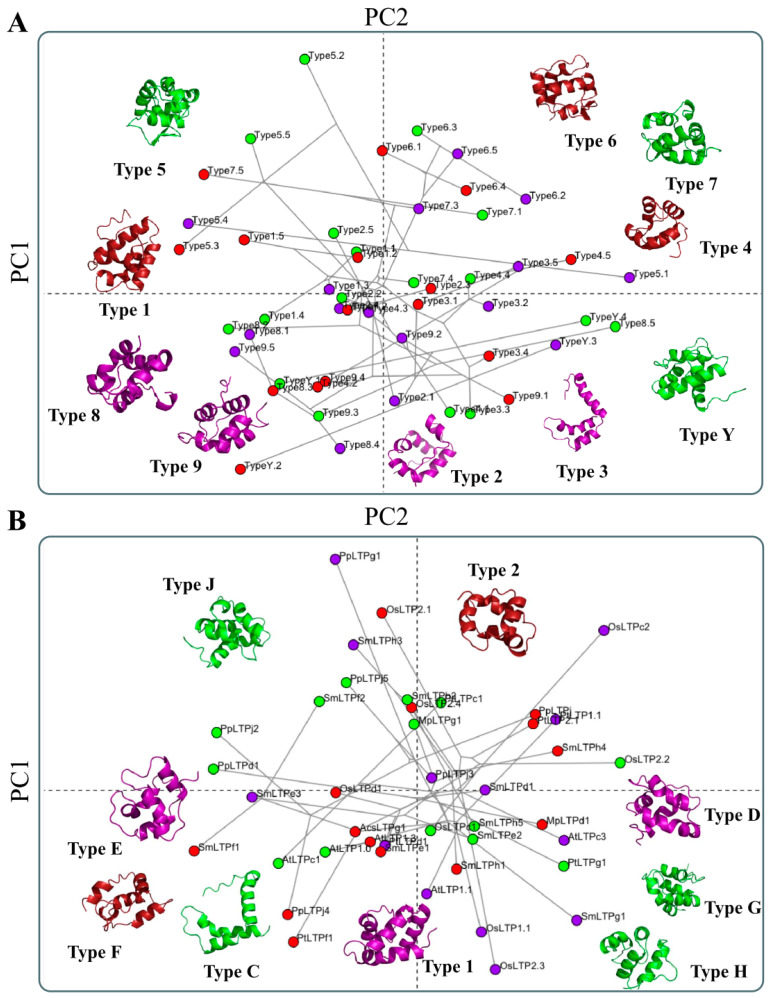
Principal components analysis (PCA) of the different nsLTP types/groups’ three-dimensional structures. (**A**) Philomorpho-spatial graph of the 49 structural theoretical models of nsLTPs classified by Boutrot et al. [17]. (**B**) Philomorpho-spatial graph of the 41 structural theoretical models of nsLTPs classified by Edstam et al. [1]. Structures are colored based on the group corresponding to the principal component (PC).

**Figure 6 antibiotics-12-00939-f006:**
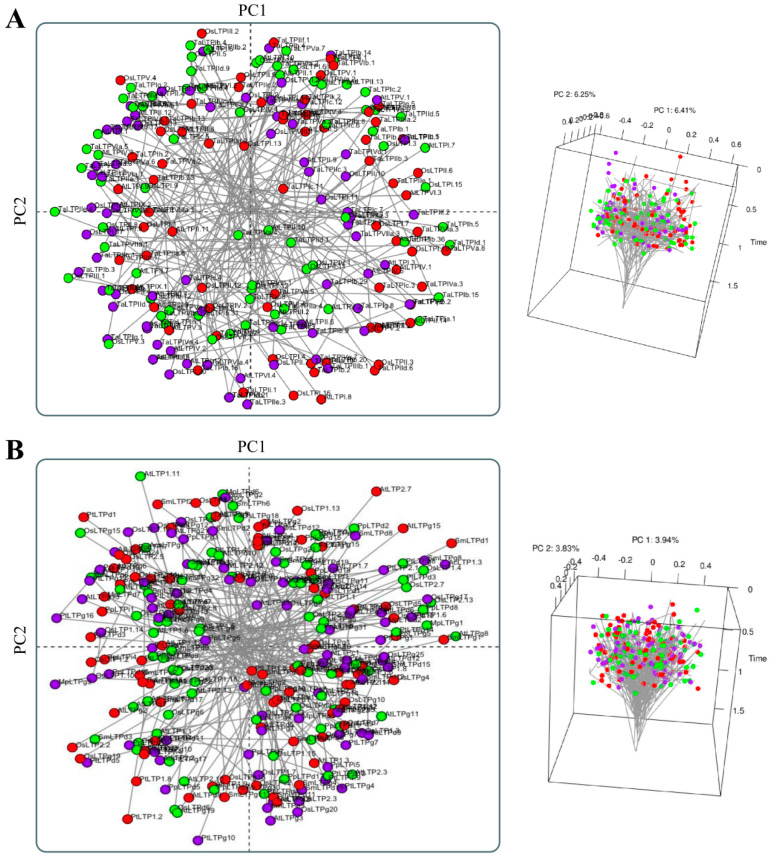
Principal components analysis (PCA) of different nsLTP types/groups. (**A**) Philomorpho-spatial chart of the amino acid physicochemical characteristics of 216 nsLTPs classified by Boutrot et al. [17]. (**B**) Philomorpho-spatial chart of the amino acid physicochemical characteristics of 263 nsLTPs classified by Edstam et al. [1]. On the left, a two-dimensional PCA representation (each dot represents an nsLTP sequence); on the right, a three-dimensional PCA representation (each dot represents an nsLTP sequence; axes are the principal components ‘1’, ‘2’, and ‘3’ of the dimensional space). Structures are colored based on the group corresponding to the principal component (PC).

**Figure 7 antibiotics-12-00939-f007:**
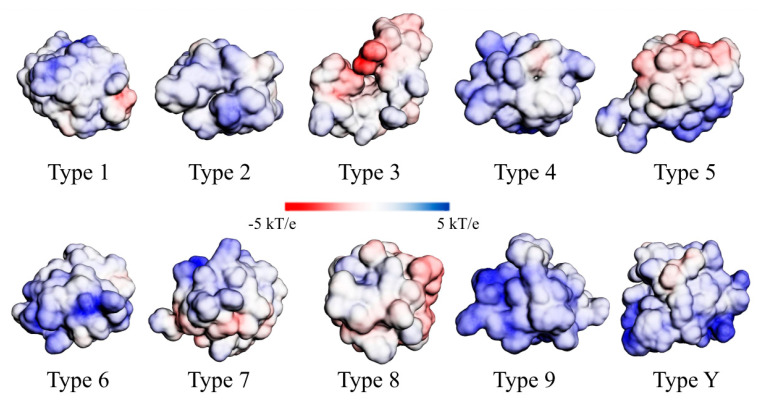
Adaptive Poisson–Boltzmann solver (APBS)-generated electrostatic surface potential plotted on the solvent accessible surface for nsLTP theoretical three-dimensional models (classification of Boutrot et al. [17]), showing the predominance of surfaces with cationic charges for most of the peptides.

**Figure 8 antibiotics-12-00939-f008:**
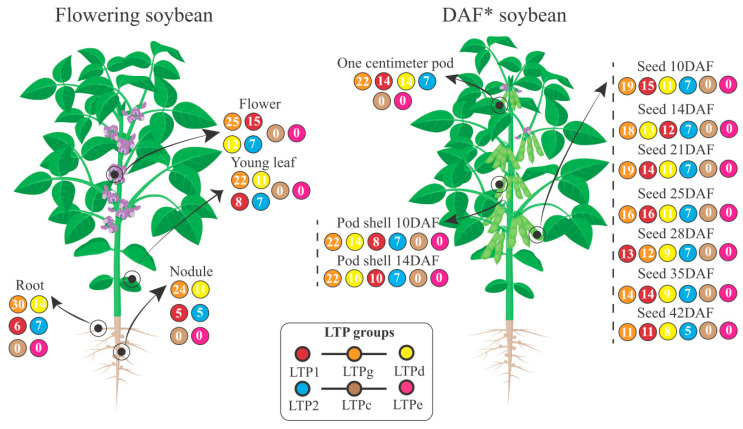
Schematic representation of soybean plants indicating the 14 treatments (tissues/stages of development) in which the GmnsLTPs (*Glycine max* nsLTPs; Edstam et al. [1] classification) expression was monitored. The box centered at the bottom of the figure indicates, through color code, the different searched GmnsLTP groups. Quantities within colored circles represent the number of expressed coding loci for the corresponding group/treatment. Legend: * DAF (days after flowering).

**Figure 9 antibiotics-12-00939-f009:**
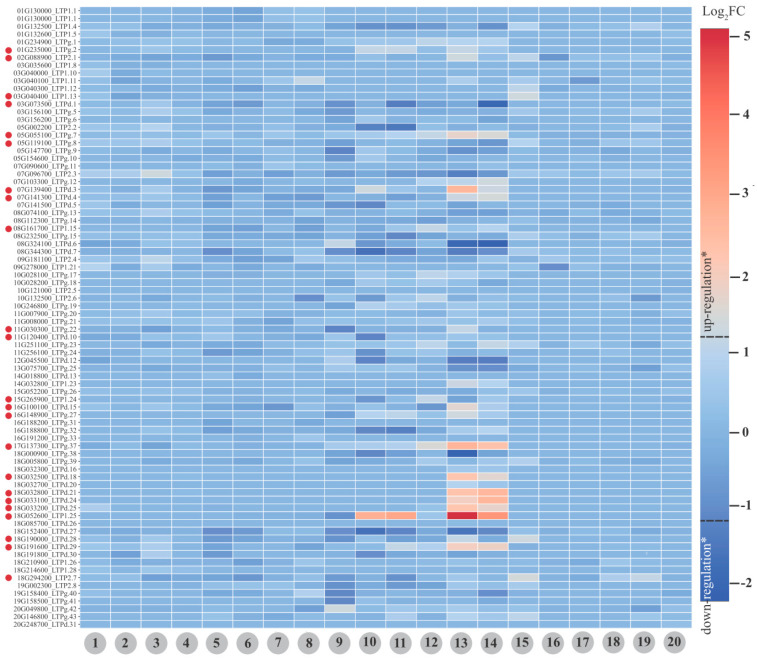
Heat map showing the GmLTP^‡^ (for Glyma 2.0 assembly) expression under different stressful conditions: 1. *‘Fusarium oxysporum* FO36 non-pathogenic’ vs. ‘control’ at ‘72 h’; 2. ‘*Fusarium oxysporum* FO36 non-pathogenic’ vs. ‘control’ at ‘96 h’; 3. ‘*Fusarium oxysporum* FO40 pathogenic’ vs. ‘control’ at ‘72 h’; 4. ‘*Fusarium oxysporum* FO40 pathogenic’ vs. ‘control’ at ‘96 h’; 5. ‘MAMP solution; LD’ vs. ‘mock; LD’; 6. ‘MAMP solution; LDX’ vs. ‘mock; LDX’; 7. ‘MAMP solution; RIL-11268’ vs. ‘mock; RIL-11268’; **8.** ‘MAMP solution; RIL-11272’ vs. ‘mock; RIL-11272’; **9.** ‘dehydration stress’ at ‘1 h’ vs. ‘control’ at ‘0 h’; 10. ‘dehydration stress’ at ‘12 h’ vs. ‘control’ at ‘0 h’; 11. ‘dehydration stress’ at ‘6 h’ vs. ‘control’ at ‘0 h’; 12. ‘salt stress’ at ‘1 h’ vs. ‘control’ at ‘0 h’; 13. ‘salt stress’ at ‘12 h’ vs. ‘control’ at ‘0 h’; 14. ‘salt stress’ at ‘6 h’ vs. ‘control’ at ‘0 h’; 15. ‘water deficit, 30% soil field capacity’ vs. ‘control, 70% soil field capacity’ at ‘Zeitgeber time 0’; 16. ‘water deficit, 30% soil field capacity’ vs. ‘control, 70% soil field capacity’ at ‘Zeitgeber time 12’; 17. ‘water deficit, 30% soil field capacity’ vs. ‘control, 70% soil field capacity’ at ‘Zeitgeber time 16’; 18. ‘water deficit, 30% soil field capacity’ vs. ‘control, 70% soil field capacity’ at ‘Zeitgeber time 20’; 19. ‘water deficit, 30% soil field capacity’ vs. ‘control, 70% soil field capacity’ at ‘Zeitgeber time 4’; 20. ‘water deficit, 30% soil field capacity’ vs. ‘control, 70% soil field capacity’ at ‘Zeitgeber time 8’. Legend: red spheres indicate up-regulated GmLTP loci (*Log_2_FC > 1 and FDR < 0.05) in at least one of the analyzed treatments; ^‡^ in chromosomal order, with abbreviated nomenclature (without the Glyma epithet), and followed by their respective nsLTP categorization.

**Table 1 antibiotics-12-00939-t001:** Some previous studies involving nsLTP mining in plant omics data, including data sources, identification strategy, amount per species, and classes retrieved.

Analyzed Species	Class	Mining Methodology	nsLTPs Amount	nsLTPs Classification	References
*Oryza sativa* (Os), *Arabidopsis thaliana* (At)	Monocot/dicot	tBLASTn/BLASTn	Os (52), At (49)	1, 2, 3, 4, 5, 6, 7, 8, and Y	Boutrot et al. [17]
*Solanum tuberosum* (St), *Solanum lycopersicum* (Sh), *Nicotiana tabacum* (Nt), *Nicotiana benthamiana* (Nb), *Capsicum annuum* (Ca), and *Petunia hybrida* (Ph)	Dicots	BLASTn	St (28), Sl (28), Nt (33), Nb (17), Ca (19), Ph (10)	1, 2, 3, 4, 5, 8, and 9	Liu et al. [21]
*Adiantum capillus-veneris* (Ac-v), *Marchantia polymorpha* (Mp), *Physcomitrella patens* (Pp), *Pinus taeda* (Pt), *Selaginella moellendorffii* (Sm), and green algae (ga)	Chlorophyta, bryophyta, dicots, and monocots	tBLASTn/BLASTn/HMM	Mp (14), Pp (40), Sm (43), Ac-v (6), Pt (40), ga (0)	1, 2, C, D, E, F, G, H, J, and K *	Edstam et al. [1]
*Lotus japonicus*	Dicot	BLAST	24	1, 2, 3, 4, 5, 8, and 9	Tapia et al. [22]
*Brassica rapa*	Dicot	BLASTp	63	1, 2, 3, 4, 5, 6, 8, 9, and 11	Li et al. [23]
*Zea mays*	Monocot	BLASTp/HMM	63	1, 2, C, D, and G	Wei e Zong et al. [5]
*Gossypium arboreum* (Ga), *Gossypium raimondii* (Gr), and *Gossypium hirsutum* (Gh)	Dicot	BLASTp	Ga (51), Gr (47), Gh (91)	1, 2, 3, 4, 5, 6, 8, and 9	Li et al. [6]
*Brassica oleracea*	Dicot	HMM/BLASp	89	1, 2, C, D, E, and G	Ji et al. [24]
*Triticum aestivum*	Monocot	BLAST Search	105	1 and 2	Hairat et al. [25]
*Triticum aestivum*	Monocot	tBLASTn	461	1, 2, C, D, and G	Kouidri et al. [26]
*Hordeum vulgare*	Dicot	BLAST Search	70	1, 2, C, D, and G	Zhang et al. [27]
*Solanum lycopersicum*	Dicot	HMM	64	1, 2, 3, 4, 10, and 11	D’Agostino et al. [28]
*Solanum tuberosum*	Dicot	BLASTp/tBLASTn/HMM	83	1, 2, 4, 5, 7, 8, 12, and 13 **	Li et al. [29]
*Triticum aestivum*	Monocot	BLASTp/HMM	330	1, 2, C, D, and G	Fang et al. [30]
*Arachis duranensis*	Dicot	HMM	64	1, 2, C, D, E, and G	Song et al. [31]
*Sesamum indicum*	Dicot	BLASTp/HMM	52	1, 2, 3, 4, 5, 6, 8, 9, and 11	Song et al. [32]
*Hordeum vulgare* (Hv) and highland barley (hb)	Dicot	BLASTp/HMM	Hv (40), hb (35)	1, 2, C, D, and G	Duo et al. [33]
*Chlamydomonas reinhardtii* (Cr), *Marchantia polymorpha* (Mp), *Physcomitrella patens* (Pp), *Selaginella moellendorffii* (Sm), *Zea mays* (Zm), *Sorghum bicolor* (Sb), *Oryza sativa* (Os), *Arabidopsis thaliana* (At), *Phaseolus vulgaris* (Pv), *Glycine max* (Gm), *Medicago truncatula* (Mt), *Trifolium pratense* (Tp), *Lotus japonicus* (Lj), *Lupinus albus* (La), and *Pisum sativum* (Ps)	Chlorophyta, bryophyta, dicots, and monocots	BLASTp	Cr (1), Mp (13), Pp (28), Sm (23), Zm (68), Sb (63), Os (73), At (82), Pv (77), Gm (120), Mt (95), Tp (85), Lj (72), La (87), Os (73)	1, 2, C, D, E, and G *	Fonseca-García et al. [34]
*Brassica napus*	Dicot	BLASTx	246	1, 2, C, D, and G	Liang et al. [35]
*Helianthus annuus*	Dicot	BLASTp/HMM	101	1, 2, 3, and 4	Vangelisti et al. [36]
*Sorghum spontaneum*	Dicot	BLAST, TBLASTN	7	1 and 2	de Oliveira Silva et al. [37]
*Brassica napus*	Dicot	HMM	238	1, 2, 3, 4, 5, 6, and 7	Xue et al. [38]

* All nsLTP groups found in the studied species pool; the nsLTP groups varied among the analyzed species. ** According to the similarity of the eight cysteine motif domains in amino acid sequences, *Solanum tuberosum* sequences were classified into two new types (12 and 13) by Li et al. [29].

**Table 2 antibiotics-12-00939-t002:** Studied species and number of recovered nsLTPs from the three applied mining approaches.

Plant Categorization			Common Name	Genome Version *	2n	Genome Size (Gb)	Mining Method **			Number of NR Loci
Higher Classification	Family	Species					BLASTp	RegEx	HMM	
Bryophyta	Marchantiaceae	*Marchantia polymorpha*	Common liverwort	Mpolymorpha_320_v3.1	18	~0.29	0	2	21	21
Pteridophyta	Pteridaceae	*Ceratopteris richardii*	Triangle waterfern	Crichardii_676_v2.1	78	~11.25	0	64	65	65
	Selaginellaceae	*Selaginella moellendorffii*	Spikemoss	Smoellendorffii_91_v1.0	16	~0.1	0	0	36	36
Gymnosperm	Cupressaceae	*Thuja plicata*	Western redcedar	Tplicata_572_v3.1	22	~12.5	4	12	112	112
Angiosperm	Malvaceae	*Gossypium hirsutum*	Cotton	Ghirsutum_527_v2.1	52	~2.43	67	54	218	218
	Asteraceae	*Lactuca sativa*	Lettuce	Lsativa_467_v5	18	~2.5	55	44	105	105
	Euphorbiaceae	*Manihot esculenta*	Cassava	Mesculenta_671_v8.1	36	~0.7	29	28	98	98
	Phrymaceae	*Mimulus guttatus*	Monkeyflower	MguttatusTOL_551_v5.0	28	~0.4	14	22	114	114
	Salicaceae	*Populus trichocarpa*	Black cottonwood	Ptrichocarpa_533_v4.1	19	~0.5	25	22	85	85
	Brassicaceae	*Sinapsis alba*	White mustard	Salba_584_v3.1	24	~0.5	23	64	189	189
	Solanaceae	*Solanum tuberosum*	Potato	Stuberosum_686_v6.1	48	~0.8	35	24	105	105
	Amaranthaceae	*Spinacia oleracea*	Spinach	Soleracea_575_Spov3	12	~0.9	6	8	43	43

* Phytozome genome version used to infer the conceptual proteome; ** number of nsLTP sequences with validated eight-cysteine domain (8CM). Legend: RegEX (regular expression); HMM (Hidden Markov Model); Gb (gigabase); NR (non-redundant).

**Table 3 antibiotics-12-00939-t003:** nsLTP quantification by categories in the 12 analyzed plant genomes.

Plant Clade	Plant Species	nsLTP Category									Total ^2^
		LTP1	LTP2	LTPd	LTPg	LTPc	Unknown 1	Unknown 2	Unknown 3	Unknown 4	
Bryophyta	Mp	1	1	10	4	0	3	0	0	2	21
Pteridophyta	Sm	0	1	12	6	1	10	0	0	6	36
	Cr	32	0	0	2	5	14	0	2	10	65
Gymnosperm	Tp	32	0	9	41	2	4	2	22	0	112
Angiosperm	So	14	0	7	17	1	0	1	3	0	43
	Pt	17	3	15	33	2	1	1	13	0	85
	Me	24	5	11	29	1	1	2	25	0	98
	Ls	26	10	7	31	1	2	10	17	1	105
	St	41	4	7	28	0	2	4	19	0	105
	Mg	40	4	10	28	2	5	10	15	0	114
	As	25	15	31	70	2	2	12	32	0	189
	Gh	30	17	34	76	5	3	5	48	0	218
	Total ^1^	282	60	153	365	22	47	47	196	19	1191

Legend: Marchantia polymorpha (Mp), Ceratopteris richardii (Cr), Selaginella moellendorffii (Sm), Thuja plicata (Tp), Glycine max (Gm), Gossypium hirsutum (Gh), Lactuca sativa (Ls), Manihot esculenta (Me), Mimulus guttatus (Mg), Populus trichocarpa (Pt), Sinapis alba (Sa), Solanum tuberosum (St), Spinacea oleracea (So), and Unk. (unknown); Total ^1^ (total of loci by category); Total ^2^ (total of loci encoding the nsLTPs in each plant genome).

## Data Availability

The data presented in this study are available in the Appendix A section.

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
