# Peer review of "From Gene to Transcript and Peptide: A Deep Overview on Non-Specific Lipid Transfer Proteins (nsLTPs)"

_antibiotics, 2023, doi:10.3390/antibiotics12050939_

Round 1

Reviewer 1 Report

Santos-Silva et al. present a comprehensive review of non-specific lipid transfer proteins (nsLTPs) in their manuscript. Apart from generally being a compendium about nsLTPs, the authors have presented a readable publication with an overall sound statistical presentation of the nsLTP paralogs in numerous plant species. 

That said, one part - 4. the structural proteomics section reveals areas of necessary improvement. Figure 5 and 6 provide the reader with principle component analysis of three-dimensional structures and types/groups of nsLTPs. 
It is debatable if either figure 5 or 6 provide any value to the reader at all, except that a PCA analysis is obviously not the right tool to address the question that is depicted by the authors. Figure 5 could end up in the supplement, whereas the 'hairball' depiction of figure 6 needs to be dropped completely out of the manuscript. Much of the text for figure 5 and 6 can be compacted in the text. 

Author Response

Dear Reviewer #1,

We would like to express our sincere gratitude for your insightful comments. We greatly appreciate the time and effort you dedicated to carefully assessing our work. We assure you that we diligently considered the suggestion provided.

Below is our response to your comment. We remain available for further clarification if needed.

Sincerely,

The authors.

‘Response to Reviewer #1’

Dear Reviewer #1,

Principal Component Analysis (PCA) is a widely used statistical technique for analyzing multidimensional data, including proteomics (Shafee et al., 2016; Shafee & Anderson, 2018). The high-dimensional space of multidimensional data is transformed through rotation and projection into a smaller set of dimensions that succinctly encapsulate the main patterns of covarying properties via PCA. This process captures essential characteristics of the sequence space in a reduced, human-understandable dimensionality, akin to observing a three-dimensional shadow of a complex, multi-dimensional object (Shafee & Anderson, 2018).

The referred technique can be instrumental in identifying trends and patterns within complex datasets. In our work, by applying PCA analysis to three-dimensional (3D) structures and physicochemical parameters of nsLTPs, we observed the absence of significant protein groups or subgroups with similar patterns, which could aid in understanding the biological functions of these proteins (Santos-Silva et al., 2021). Furthermore, the PCA analysis categorically suggests that 3D structure and physicochemical parameters cannot be considered independently as criterions for classification due to its 3D structural conservation and physicochemical parameters variation. Interestingly, despite the significant diversity of amino acid residues in the primary structure of the analyzed nsLTPs, their respective 3D structures exhibited pronounced conservation. Therefore, we emphasize that the inclusion of Figures ‘5’ and ‘6’ in the main body of the manuscript can provide valuable insights to readers interested in understanding the nsLTPs structural analysis and properties, and the current intricacies of classifying this protein group. Our PCA analysis was inspired by the Shafee & Anderson (2018) work, particularly in Figures “1”, “2”, and “3”. 

With the above, we trust that your question has been clarified accordingly, and we hope you will accept our arguments, which are based on the scientific literature. We remain at your disposal for any further clarification and we thank you once again for your meticulous reading of our work.

Sincerely,

The authors.

Cited references:

Shafee, T. M., Lay, F. T., Hulett, M. D., & Anderson, M. A. (2016). The defensins consist of two independent, convergent protein superfamilies. Molecular biology and evolution, 33(9), 2345-2356.

Shafee, T., & Anderson, M. A. (2018). A quantitative map of protein sequence space for the cis-defensin superfamily. Bioinformatics, 35(5), 743-752.

Santos-Silva, C. A., Vilela, L. M. B., de Oliveira-Silva, R. L., da Silva, J. B., Machado, A. R., Bezerra-Neto, J. P., ... & Benko-Iseppon, A. M. (2021). Cassava (Manihot esculenta) defensins: prospection, structural analysis and tissue-specific expression under biotic/abiotic stresses. Biochimie, 186, 1-12.

Reviewer 2 Report

Non-specific lipid transfer proteins (nsLTPs) constitute a vast and diverse gene family that is widely distributed across various plant genomes. However, due to this gene family's intricate and heterogeneous nature, developing a standardized classification system for nsLTPs has remained a challenge, resulting in a lack of consensus among researchers.

The comprehensive review conducted by Santos-Silva and colleagues has made significant strides in consolidating and presenting relevant information on nsLTPs from multiple perspectives, including omics studies, evolutionary analysis, and structural proteomics. Notably, their emphasis on the classification of nsLTPs provides valuable insights into these proteins' diverse characteristics and functional attributes.

Moreover, the authors conducted an insightful case study focusing on soybean, encompassing a comprehensive transcriptional analysis. This analysis encompasses the examination of baseline gene expressions across different tissues and developmental stages, as well as the investigation of differential gene expressions under various biotic and abiotic stresses. By elucidating the spatiotemporal regulation of nsLTPs in soybean, this case study offers significant contributions to our understanding of the dynamic behavior of these proteins.

In summary, Santos-Silva and colleagues' work represents a critical review and meta-analysis that effectively consolidates high-quality information on nsLTPs into a single source. Through the integration of diverse research findings and the presentation of original results, this review not only sheds light on previously unexplored aspects of nsLTPs but also contributes to the broader understanding of this important gene family.

Author Response

Dear Reviewer #2,

We would like to express our sincere gratitude for your thorough review of our manuscript.

Sincerely,

The authors.

Reviewer 3 Report

The manuscript entitled „From Gene to Transcript and Peptide: A Deep Overview on Non-specific Lipid Transfer Proteins (nsLTPs)” by dos Santos-Silva et al. provides a meta-analysis of snLTPs in various plant genomes including plants which have not been analysed before combined with proteomics and structural analysis. The study provides a very comprehensive overview and detailed investigations.

 There are only a few comments:

 -        It is recommended to show amino-acid sequence alignments of the representative snLTPs group members in the Introduction.

 -        Table 1 – the legend should refer to the abbreviations used in column “nsLTP amount”. Alternatively, the abbreviation could be shown in bracket after the species (e.g. Oryza sativa (Os)) in column “Analyzed species”.

-        Abbreviations for plant-specific nsLTPs should be shown – page 6 second paragraph “…nsLTP for potato (Solanum tuberosum; StnsLTP)…”

-        Page 6, paragraph 3 – contains three times AdLTP instead of AdnsLTP

-        Page 9, Paragraph 2: S. moellendorfii is named twice “…from the hypothetical proteomes of C. richardii, S. moellendorffii and S. moellendorffii (Table 2).”

Author Response

Dear Reviewer #3,

We would like to extend our heartfelt appreciation for the meticulous review of our manuscript. We assure you that we diligently addressed each of the suggestions provided, and we are confident that they will contribute to enhancing the quality of our research. Once again, thank you for your invaluable contribution to the advancement of our study.  Below, in the ‘'Response to Reviewer’ item, we present a point-by-point explanation of your inquires.

Sincerely,

The authors.

‘Response to Reviewer #3’

  1. It is recommended to show amino-acid sequence alignments of the representative snLTPs group members in the Introduction.

Authors’ response: Dear Reviewer #3, thank you for your suggestion. The ‘Boutrot et al. (2008)’ and ‘Eldstan et al. (2011)’ works, which serve as significant references for the development of a more suitable classification system for the protein group under study, provide sequence groups either within the main body of the text or in the supplementary materials section. Hence, we believe there is no need to include such content in the present manuscript. We appreciate your understanding and remain available for any further clarification.

  1. Table 1 – the legend should refer to the abbreviations used in column “nsLTP amount”. Alternatively, the abbreviation could be shown in bracket after the species (e.g. Oryza sativa (Os)) in column “Analyzed species”.

Authors’ response: Dear Reviewer #3, we have included the abbreviations in the “Analyzed species” column, as you suggested. All the changes can be viewed by activating the “Track Changes” function in the Word file. Thank you for your recommendation.

  1. Abbreviations for plant-specific nsLTPs should be shown – page 6 second paragraph “…nsLTP for potato (Solanum tuberosum; StnsLTP)…”

Authors’ response: Dear Reviewer #3, we have included the abbreviations for plant-specific nsLTPs, as you suggested. All the changes can be viewed by activating the “Track Changes” function in the Word file. Thank you for your recommendation.

  1. Page 6, paragraph 3 – contains three times AdLTP instead of AdnsLTP.

Authors’ response: Dear Reviewer #3, The requested correction has been made. Thank you for bringing it to our attention. All the changes can be viewed by activating the “Track Changes” function in the Word file.

  1. Page 9, Paragraph 2: S. moellendorfii is named twice “…from the hypothetical proteomes of C. richardii, S. moellendorffii and S. moellendorffii (Table 2).”

Authors’ response: Dear Reviewer #3, The requested correction has been made. The second term 'S. moellendorffii' has been replaced with 'M. polymorpha'. Thank you for bringing it to our attention. All the changes can be viewed by activating the “Track Changes” function in the Word file.